Childhood and adult socioeconomic position interact to predict health in mid life in a cohort of British women

Nettle Daniel daniel.nettle@ncl.ac.uk
Bateson Melissa
Centre for Behaviour and Evolution & Institute of Neuroscience, Newcastle University, Newcastle , UK
Cooper Richard
Electronic publication date: 2017 Jun 29
Publication date: 2017
Volume: 5
Electronic Location ID: e3528
Received 2017 Mar 29; Accepted 2017 Jun 10
Copyright: ©2017 Nettle and Bateson
Copyright year: 2017
Copyright holder: Nettle and Bateson
License: This is an open access article distributed under the terms of the Creative Commons Attribution License, which permits unrestricted use, distribution, reproduction and adaptation in any medium and for any purpose provided that it is properly attributed. For attribution, the original author(s), title, publication source (PeerJ) and either DOI or URL of the article must be cited.
License URL: https://creativecommons.org/licenses/by/4.0/

Keywords: Socioeconomic position, Health, Health inequalities, Silver spoon, Mismatch, Childhood

Funding: European Research Council (ERC), European Union, Horizon 2020 research and innovation programme AdG 666666 COMSTAR This project has received funding from the European Research Council (ERC) under the European Union’s Horizon 2020 research and innovation programme (grant agreement no. AdG 666666 COMSTAR). The funders had no role in study design, data collection and analysis, decision to publish, or preparation of the manuscript.

==============================
Background

Low childhood socioeconomic position (cSEP) is associated with poorer adult health, even after adult socioeconomic position (aSEP) is adjusted for. However, whether cSEP and aSEP combine additively or non-additively in predicting adult health is less well studied. Some evidence suggests that the combination of low cSEP and low aSEP is associated with worse health than would be predicted from the sum of their individual effects.

Methods

Using data from female members of the British National Child Development Study cohort, we developed continuous quantitative measures of aSEP and cSEP, and used these to predict self-rated health at ages 23, 33, and 42.

Results

Lower aSEP predicted poorer heath at all ages. Lower cSEP predicted poorer health at all ages, even after adjustment for aSEP, but the direct effects of cSEP were substantially weaker than those of aSEP. At age 23, the effects of cSEP and aSEP were additive. At ages 33 and 42, cSEP and aSEP interacted, such that the effects of low aSEP on health were more negative if cSEP had also been low.

Conclusions

As women age, aSEP and cSEP may affect their health interactively. High cSEP, by providing a good start in life, may be partially protective against later negative impacts of low aSEP. We relate this to the extended ‘silver spoon’ principle recently documented in a non-human species.

Introduction

Evidence has accumulated that lower childhood socioeconomic position (cSEP) is associated with poorer health many years later in adulthood (Elo & Preston, 1992; Galobardes, Lynch & Smith, 2004; Pollitt, Rose & Kaufman, 2005; Pollitt et al., 2007; Cohen et al., 2010; Zimmer, Hanson & Smith, 2016). This could simply be because low cSEP leads to low adult socioeconomic position (aSEP), with low aSEP having direct negative effects on health (Marmot et al., 2001). If this were the case, associations between cSEP and adult health would disappear once aSEP was controlled for. However, recent evidence suggests that this is often not the case: controlling for aSEP tends to attenuate but not eliminate the associations between cSEP and adult health (Pollitt, Rose & Kaufman, 2005; Pollitt et al., 2007; Cohen et al., 2010; Zimmer, Hanson & Smith, 2016). This suggests there may be lasting direct impacts of exposure to low cSEP that produce health consequences years later.

There are a number of different models of how exposures combine across the life-course to influence adult health. For example, early-life exposures may have specific somatic consequences that are initially latent and only surface in morbidity years later (“the critical period” or “latency” model, Kuh et al., 2003). Alternatively, it may be that what is important is simply the amount of low SEP that one has been exposed to over the course of life (the “accumulation model”, Kuh et al., 2003). In this case, individuals experiencing both low cSEP and low aSEP have simply been exposed to more low SEP overall than individuals experiencing one but not the other.

Neither the latency model nor the accumulation model specifies whether effects of aSEP and cSEP will be additive or interactive. Both models would be potentially compatible with finding that the negative effect of low cSEP and low aSEP combined is simply the sum of the effects of each considered separately. Equally, both models could accommodate a situation where the combined impact of low cSEP and low aSEP is worse than the sum of their isolated impacts. For example, the accumulation model could generate non-additive interactions between cSEP and aSEP if the mapping between the total amount of low SEP experienced over the life-course, and health, were a non-linear function. Interactions between cSEP and aSEP have been examined much less frequently than the question of whether there is an effect of cSEP on adult health independently of aSEP. One study found that cSEP and aSEP combined additively rather than interactively (Nandi et al., 2013), whilst two others found evidence that low cSEP and low aSEP were worse for health when combined than the sum of their effects in isolation (Wamala, Lynch & Kaplan, 2001; Claussen, Davey Smith & Thelle, 2003).

Modelling potential interactions has been difficult as many previous studies employed only crude, often categorical, measures of SEP (Cohen et al., 2010). Moreover, cSEP and aSEP are often measured on incomparable scales. Thus, although the consensus is that the direct effects of cSEP are weaker than the direct effects of low aSEP (Pollitt et al., 2007), it has been hard to compare these quantitatively. Here, we sought to develop continuous single measures of cSEP and aSEP for a longitudinally-studied cohort of British women. Although SEP has different components, which may have differential associations with health, they tend to be correlated and hence can be combined into single measures for purposes such as the present one. We then modelled the associations of cSEP and aSEP with self-rated health at three ages, 23, 33 and 42. Self-rated health is a simple outcome measure that has been shown to correlate well with more objective markers, and to prospectively predict future morbidity and mortality (DeSalvo et al., 2006; Christian et al., 2013; Cabrero-García & Juliá-Sanchis, 2014). For each age point, we first sought to establish whether there was an association between cSEP and health once aSEP was adjusted for, and to compare the strength of the cSEP-health association to that of the aSEP-health association. We next tested whether an interactive model fitted the data better than an additive one, and if so, probed the nature of the interaction.

Materials & Methods

Study cohort

The National Child Development Study (NCDS) is an ongoing longitudinal cohort study of all children born in Britain between March 3rd and March 9th 1958 (Power & Elliott, 2006). As the present study arose from a previous one on timing of pregnancy (Nettle, Coall & Dickins, 2011), only female cohort members are included. The female cohort size at birth was 8959. Some cohort members miss each of the periodic surveys, and others have been permanently lost to follow-up. The number of women with at least one rating of adult health in the present study was 7,232 (6,266 ratings at age 23 [Sweep 4]; 5,727 at age 33 [Sweep 5]; 5,773 at age 42 [Sweep 6]).

Data accessibility

Researchers may apply to access the NCDS data through the UK Data Service (www.ukdataservice.ac.uk). The original NCDS variable labels for the variables included or derived in this study are listed in the Supplemental Information, section 1.

Characterising aSEP and cSEP

We summarised cSEP by performing a principal components analysis on three childhood variables: father’s social class at the child’s birth; mother’s age at leaving education; and the proportion of fathers in the child’s school class at age 7 (Sweep 1) who were from a professional occupation (see Supplemental Information for full details of the PCA). As well as producing a single summary measure, principal components analysis has the advantage of producing a standardized variable with better distributional characteristics for modelling than each of the variables that contributes to it.

We summarised SEP over the course of adulthood up until the time of each of our three health measures by similar PCA methods. At age 23, the aSEP variable was from a PCA of social class and highest educational qualifications at age 23. The aSEP variable at age 33 was from a PCA that also added in social class and highest educational qualifications at age 33; and at 42, the aSEP variable was from a PCA that also added in social class at age 42 and household income at age 42. Thus, our variables effectively captured cumulative exposure to SEP across increasingly large intervals of adult life. Social class at all time points was measured using the Registrar General’s occupational classification, a standard classification for the period in British social statistics. The correlations of the three aSEP measures with one another were very high (all rs > 0.88), whilst the correlations of aSEP to cSEP for individuals who had at least one health measure were: 0.35 (cSEP–aSEP aged 23); 0.37 (cSEP–aSEP aged 33); and 0.37 (cSEP–aSEP aged 42).

Self-rated health

Self-rated health was reported using a single question at interviews in 1981, 1991 and 2000 (ages 23, 33 and 42). Responses were on a four- point scale (poor/fair/good/excellent). Given the large sample size and ordered structure of the responses, we here treated the health scale as a continuous variable. The distribution and homoscedasticity of residuals were examined for each model and justified this approach.

Analytical approach

For each age point, we adopted the analytical approach specified below, following standard guidelines for model selection (Symonds & Moussalli, 2010). We fitted five possible models of the data, with the predictors as follows: 1. Intercept only (i.e., a null model); 2. Intercept and aSEP; 3. Intercept and cSEP; 4. Intercept plus aSEP and cSEP additively; 5. Intercept plus aSEP, cSEP and their interaction. These models were then compared by AICc, an information-theoretic measure of model fit, using the R package ‘AICcmodavg’ (Mazerolle, 2015). This procedure gives each of the five models a weight, which can be interpreted as the strength of support for that model being the best model of the data in the set (Symonds & Moussalli, 2010). We explored the interactions observed using simple slopes analysis (Preacher, Curran & Bauer, 2006). This uses the estimated model coefficients to calculate the association of aSEP with health when cSEP takes different values; in the present case, one standard deviation below the mean cSEP, at the mean cSEP, and one standard deviation above mean cSEP.

Results

Table 1 shows the AICc values given to each model at each age point, and the corresponding AICc weights. At all ages, most support was given to models including both aSEP and cSEP rather than those omitting one or both of those variables (total weight for such models 0.94 at age 23; 1.00 at age 33, and 1.00 at age 42). At age 23, the additive model 4 had a lower AICc and was given higher weight than the interactive model 5, whereas at 33 and 42, the interactive model 5 had lower AICc and higher weight. This suggests that both aSEP and cSEP are important for predicting health, that their effects are approximately additive at 23, but there is an interaction between them that has emerged by age 33. This inference is supported by the fact that the confidence interval for the parameter estimate of the interaction term between cSEP and aSEP in model 5 includes zero at age 23 (B =  − 0.011, 95% CI [−0.031–0.009], but does not include zero at age 33 (B =  − 0.020, 95% CI [−0.040 to −0.001] or age 42 (B =  − 0.026, 95% CI [−0.045 to −0.006]; Fig. 1C).

Table 1 Model fit indices for the five possible models of self-rated health at each age point.

Model	Age 23	Age 33	Age 42	
	ΔAICc	Weight	ΔAICc	Weight	ΔAICc	Weight	
Model 1: Null	163.22	0.00	284.53	0.00	282.98	0.00	
Model 2: aSEP only	4.38	0.06	16.18	0.00	19.59	0.00	
Model 3: cSEP only	118.21	0.00	184.16	0.00	188.33	0.00	
Model 4: aSEP + cSEP additive	0.00	0.56	2.33	0.24	4.71	0.09	
Model 5: aSEP * cSEP interactive	0.80	0.38	0.00	0.76	0.00	0.91	
Notes.

ΔAICc: Difference in AICc between the listed model and the best (lowest AICc) model in the set, hence 0 for the best model itself. Weight: AICc weight, interpreted as the strength of support for each of the models being the best model of the data in the set.

Figure 1 Estimates of the association between SEP measures and health at each age point (standardised β coefficients).

(A) Adult SEP, from the unadjusted model 2 (open circles), and from model 4 which also includes childhood SEP (filled squares and confidence intervals). (B) Childhood SEP, from the unadjusted model 3 (open circles) and from model 4 which also includes adult SEP (filled squares and confidence intervals). (C) The interaction term between adult and childhood SEP, from model 5.

Figure 1 visualizes the strengths of associations between the predictors and health at each age. aSEP is associated with health at all age points, and the standardised parameter estimate is little changed by adjusting for cSEP (Fig. 1A). The association of cSEP to health is attenuated by around 50% when aSEP is included in the model compared to the unadjusted association (Fig. 1B). However, its 95% confidence interval still does not include zero at any age point. When both aSEP and cSEP are included in the same model, the associations of cSEP to adult health are only around one third the strength of those of aSEP to adult health.

To explore interactions between aSEP and cSEP that are apparently present at ages 33 and 42, we used simple slopes analysis to calculate the predicted value of health across the range of adult SEP, for low cSEP (1 s.d. below mean), average (mean), and high (1 s.d. above mean; Fig. 2). At both ages, predicted health is lower when aSEP is lower for all values of cSEP, but the negative effects of low aSEP are stronger the lower cSEP was. A corollary is that cSEP becomes unimportant for health if aSEP is sufficiently high.

Figure 2 Predicted self-rated health: (A) at age 33; and (B) at age 42.

Predicted health is shown across the range of adult SEP, for three different values of childhood SEP (cSEP): one standard deviation below the mean, the mean, and one standard deviation above the mean. Predictions are from statistical model 5.

Discussion

By creating single, continuous standardised measures of aSEP and cSEP for a cohort of British women, we were able to examine their relative predictive power for health in adulthood, and determine whether they interacted. We found that both aSEP and cSEP were related to health. We also showed that by ages 33 and 42, they interacted, such that low cSEP makes the negative effects of low aSEP on health stronger. The result can be equivalently described as higher aSEP protecting the individual from the negative effects of lower cSEP.

Our findings confirmed previous evidence that cSEP is related to health many years later (Elo & Preston, 1992; Galobardes, Lynch & Smith, 2004; Pollitt, Rose & Kaufman, 2005; Pollitt et al., 2007; Cohen et al., 2010; Zimmer, Hanson & Smith, 2016). We concur with other studies that this is not entirely due to low cSEP leading to low aSEP. We did find that cSEP predicted aSEP moderately strongly, indicating intergenerational persistence of socioeconomic position in this cohort. Such persistence has been observed before, although it has been shown to be somewhat less strong for Britons born in 1958 than for more recent generations (Blanden, Gregg & Macmillan, 2007). However, although adjusting for aSEP attenuated the association of cSEP to health, it did not reduce it to zero (Cohen et al., 2010). In fact, our modelling approach allows us to be quite specific: about half of the bivariate association between cSEP and adult health is due to the continuity of cSEP into aSEP; the other half is therefore not. The associations of cSEP to adult health were considerably weaker than those of aSEP to adult health, again confirming previous findings (Pollitt et al., 2007). Again, our modelling approach method allowed us to quantify how much weaker: the effect of cSEP on adult health is just under one third as strong as the effect of aSEP in a mutually-adjusted model. The association of cSEP to adult health shows no sign of weakening with age: on the contrary, the parameter estimate for cSEP was larger at age 42 than at age 23.

The mechanisms producing the interaction we found are not known in detail. Lower cSEP is likely to expose individuals to poorer diet, environmental conditions and exposures, and more psychosocial disruption and stress. Presumably, such conditions limit individuals’ investment in somatic capacities and structures at key points in development (cf. Wells, 2010). These limitations are exposed with age and produce increased morbidity. In addition, there may be behavioural pathways. For example, cSEP has been shown to be an independent predictor of alcohol-related disorders (Gauffin, Hemmingsson & Hjern, 2013), and persistent smoking in women (Jefferis et al., 2004).

Our results concur with the predictions of “latency” and “accumulation” models from life-course epidemiology, in that both of these models predict later direct health consequences of adversities experienced early in the life-course. However, neither of these models, without further specification, predicts whether earlier and later adversities will be additive or interactive in their effects. Our results suggest that they are additive in young adulthood, but become interactive as individuals age, so that later in mid-life, the combined impacts of low cSEP and aSEP are greater than their sum. We note that our findings are congruent with recent results from zebra finches (Briga et al., 2017). Briga et al. (2017) found that low food availability in adulthood had negative effects on life-span for individuals who had experienced harsh developmental conditions, whereas individuals who had experienced benign developmental conditions were protected. This is equivalent to the pattern of results shown in our Fig. 2. Briga et al. (2017) interpret their results by extending the ‘silver spoon’ principle widely discussed in evolutionary ecology (Grafen, 1988), whereby good conditions during development provide a general advantage in survival and reproductive success. Briga’s et al. (2017) results represent an extension of this principle since the advantage manifests particularly where the adult environment is harsh–effectively, what the ‘silver spoon’ provides is greater robustness against adult adversity. Our results additionally suggest that this becomes particularly true as individuals age, given that the interaction between cSEP and aSEP became stronger across the three age points.

We note that this extended ‘silver spoon’ principle stands in contrast to the ‘match-mismatch’ style of reasoning popular in discussions of the developmental origins of health and disease. According to the ‘match-mismatch’ framework, experience in early life serves as a cue of the likely conditions in the adult environment, and allows the individual to prepare phenotypically for them (Bateson et al., 2004; Gluckman, Hanson & Beedle, 2007; Hanson & Gluckman, 2014). Health problems arise if this cue turns out to be invalid, i.e., if the adult environment is ‘mismatched’ from the childhood one. If this principle applied, low cSEP should be relatively beneficial for adult health as long as aSEP is also low, whereas high cSEP would be beneficial as long as aSEP turns out to be high. This would produce an interaction between cSEP and aSEP in predicting adult health, but the direction of the interaction would be opposite to that shown in Fig. 2. This suggests that the ‘match-mismatch’ principle is not applicable to childhood experience of socioeconomic conditions in developed populations. More generally, where match-mismatch ideas have been tested on early-life conditions in human datasets, their predictions have not been supported; the overwhelming signal appears to be that early-life adversity is always bad for adult health (Hayward & Lummaa, 2013; Hayward, Rickard & Lummaa, 2013).

Our study was limited in only considering a single self-report outcome—self-rated health—with no other markers or information on what the sources of ill-health were. Global self-rated health has been found in previous studies to correlate well with more objective markers, and to predict future mortality (DeSalvo et al., 2006; Christian et al., 2013; Cabrero-García & Juliá-Sanchis, 2014). Our design precluded separating the effects of different components of SEP, such as income and education. This was deliberate, since testing interactions between cSEP and aSEP requires creating simple portmanteau measures of each. Although we were able to document associations between aSEP and health, we cannot make strong claims about the direction of causality in this association. Whilst low SEP is likely to have a negative effect on health, poor health may also lead to downward social drift (Mulatu & Schooler, 2002). Nonetheless, any causal impact of aSEP on health will be partialled out by controlling for aSEP, as we did. Thus, if there is an association of cSEP to adult health that survives controlling for aSEP, then the inference that cSEP affects adult health more directly than just by leading to lower aSEP seems a safe one, regardless of the direction of causality in the aSEP-health relationship. Reverse causality in the cSEP-adult health association seems unlikely: there is no obvious way, for example, for a woman’s health to influence her father’s social class at the time of her birth.

Our results underline the potential detrimental effects of family poverty and deprivation on health. As well as any immediate effects in childhood, there are health sequelae that manifest many decades later with ageing. As we have shown here, socioeconomic deprivation in childhood may reduce resilience to adverse conditions experienced in adulthood. A more positive, though entirely equivalent, way of stating this conclusion is that an affluent adult environment can mitigate or even eliminate the adverse effects of childhood socioeconomic deprivation on health.

Supplemental Information

Supplemental Information 1 Supplementary Information

Click here for additional data file.

Additional Information and Declarations

Competing Interests

Author Contributions

Data Availability

The authors declare there are no competing interests.

Daniel Nettle conceived and designed the experiments, performed the experiments, analyzed the data, contributed reagents/materials/analysis tools, wrote the paper, prepared figures and/or tables, reviewed drafts of the paper.

Melissa Bateson conceived and designed the experiments, contributed reagents/materials/analysis tools, reviewed drafts of the paper.

The following information was supplied regarding data availability:

The cohort dataset we use belongs to the Centre for Longitudinal Studies (www.cls.ioe.ac.uk) and we use it under registration from the UK Data Service (www.ukdataservice.ac.uk). We are not permitted to share the raw data freely (please see https://discover.ukdataservice.ac.uk/catalogue/?sn=5565&type=Data%20catalogue for the access requirements).

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
