# Peer review of "Childhood and adult socioeconomic position interact to predict health in mid life in a cohort of British women"

_PeerJ, doi:10.7717/peerj.3528_

## Round 0.1 · original submission · Major Revisions

The question under examination is important and deserves further study. As you can see, 1 reviewer had some difficulty with the silver spoon construct, which I tend to agree with. You are free to keep it of course, but might consider offering other possible - more general - explanatory frameworks. And only 1 reviewer felt "major revisions" were needed, so that was not a consensus.

·

Basic reporting

The report is clear and well structured. However, I have a few comments below:
(1) I find table 1 challenging to read, given that the models are not in the same order across all ages, but are ordered by AICc. The authors may consider restructuring the table so that it’s easier to read. My suggestion would be to do a wider table where the order of the 5 models at each age point can be seen side by side.
(2) Hanson & Beedle 2007 Reference (L56) seems to be missing from the reference list. So is Hanson&Smith 2016 (L164). Please make sure all the other references are included.
(3) To guide the reader, the authors should provide guidance on how high/low their cSEP/aSEP correlation is. Is it high or low compared to other UK studies? How about compared to other countries?

Experimental design

The basic experimental design is straightforward and looks rigorous, although I a specific methodological comment:

(1) The authors base their conclusion that aSEP and cSEP are only additive at age 23 based on AICc. While the limitations of information criteria, in general, could be further discussed in the limitations section, I disagree with the use of information criteria to test this hypothesis. I think the authors already have at their disposal a hypothesis test that is much more straightforward: is the interaction term between aSEP and cSEP significantly different from 0? This is shown in Figure 1. Given my comment above about Table 1, I think the authors may consider simplifying their analysis and not using information criteria at all to make inferences about a specific coefficient.

Validity of the findings

(1) My main concern for this manuscript is related to reverse causation:
a. The authors measure childhood SEP at birth (father’s social class, and mother’s age at leaving education, which should be time-fixed) and at age 7.
b. The authors measure adult SEP at ages 23 (social class, education), 33 (social class, education) and 42 (social class, income). This means that the entire measure of aSEP is a composite of SEP at ages 23, 33 and 42.
c. The authors then proceed to test their hypotheses by regressing self-rated health at ages 23, 33 and 42 on cSEP (birth + age 7) and/or aSEP (ages 23+33+42).
d. This means that for some of the analysis (e.g., the regression of self rated health at ages 23 or 33 which include aSEP) the authors are including measures of SEP in the future (age 42) on past health (23 or 33).
e. I am severely concerned that reverse causation may be at play here. Sicker people at age 23/33 can have lower SEP (income or social class in this case) at age 42. The second concern is that this potential sickness at age 23/33 could be caused by cSEP.
f. I believe the authors need to provide a graphical diagram (for example, a DAG) with their hypothesized causal relationships to make this picture clearer.
g. The manuscript would also benefit from measuring aSEP in a way that does not open itself to reverse causation by design. For example, use only measures available at ages 23 and 33 and regress self-rated health at age 42 on such measures.
h. This should also be discussed in the limitations section, as the literature on reverse causation and SES/health is abundant (see Oakes reviews on the topic).
Previous studies have examined some of the challenges related to aSEP/cSEP with concurrent adjustment. For example see https://www.ncbi.nlm.nih.gov/pubmed/22317806

(2) The second concern is related to the phenomenon of stickiness: since social mobility is low (which it is in the UK, not so low as the US but still low compared to other European countries), cSES affects aSES, and therefore any analysis that accounts for both needs to be very careful in its interpretation. We’ve written about this elsewhere (doi: 10.1016/j.socscimed.2016.01.001)

Additional comments

This study aims to understand the independent association of childhood and adult socioeconomic position on adult health. The authors outline two hypothesis and use a consistent measurement of SES to disentangle this relationship. Other than the comments above (re: reporting, experimental design and validity) I have a few other comments below:

(1) I am having difficulty understanding the match-mismatch model as applied to SES. The authors state that “In fact, high aSEP might even be bad for health if cSEP had been low, since the individual would be phenotypically prepared for a harsh environment, but actually experiences a benign one”. I believe this would apply well to societies in transition (i.e., growing in hardship with food insecurity and getting to adulthood with plenty of caloric intake), but current Western economies (like the one the authors use to test their hypothesis) have very few risk factors that are more prevalent in people with higher SES. I can see how a downwards SES trajectory may be especially harmful, but I think the authors need to elaborate more on how an upwards trajectory can be harmful. Maybe, I suggest, try to find studies that apply the match-mismatch theory to SES (instead of ecology). They discuss some of this in the discussion, but I am still struggling to find plausibility in the match-mismatch theory for SES.

(2) The authors may consider discussing in the intro how different is (or is not) the silver-spoon framework from a lifecourse/cumulative effects framework that the authors also mention. Some of the references the author use in the discussion to support the silver spoon hypothesis actually come from the lifecourse/cumulative effects literature.

Reviewer 2 ·

Basic reporting

• The paper is clearly written and specifies two clear hypothesis related to trajectories of life course SEP and adult self-rated health at ages 23, 33, and 42 in a sample of British women.
• The tables and figures are easy to read, and I appreciate a table of AICs to justify the model building strategy. However, some of the presentation was confusing and difficult to interpret. For example, the forest plot of an interaction term in Figure 1c don’t really make sense to me, although the slope figures in Figure 2 were helpful. I would suggest making the forest plots for adult SEP in Figure 1c set at cSEP levels of -1, 0, and 1, as was done in Figure 2.

Experimental design

• This is your basic life course design with health measures captured in young to middle adulthood. It is a cohort study with 3 separate measures of the outcome.
• The authors could have looked at a cumulative measure of SEP and health or trajectories of SEP across the full life course and health, or even trajectories of SEP against trajectories of health. But they treat the outcomes one at a time.
• Relates to this, there is some concern about adjusting for adults SEP to arrive at an independent effect of childhood SEP, as some of the effect of the latter will operate via the former. This is clearly seen in the attenuation of childhood SEP effects in figure 1b when adjusted for adult SEP.

Validity of the findings

• Many of my concerns below are about interpretation; apart from the use of a one-at-a-time approach, I have no issue with the method used or concerns about the data.
• The authors have made fairly strong claims about their results supporting a “silver spoon” theory, that is, that childhood SEP serves as a buffer against whatever comes in adulthood (as opposed to a phenotypic adaptation occurring in childhood, allowing for low SEP in childhood buffering against a “matched” low SEP in adulthood). Couldn’t these results also be consistent with a critical (adult) period model of SEP? Or a cumulative model? (see next comment)
• The magnitude of the adult coefficients and the negative interaction terms for ages 33 and 42 could also be interpreted as meaning that it is adult SEP that really matters, although it matters a bit less if the adult was low cSEP. For example, using standardized measures of cSEP and aSEP at age 33, I think you would get the following standardized predicted health measures:
cSEP aSEP Predicted Health
Age 33 -2 (low) -2 (low) -0.52
Age 33 +2 (high) -2 (low) -0.11
Age 33 -2 (low) +2 (high) +0.29
Age 33 +2 (high) +2 (high) +0.38
While they certainly do not support the match/mismatch hypothesis, they are not a clear indicator of the ‘extended’ silver spoon hypothesis either, which sounds a lot like a cumulative model. In other words, the more low SEP hits one gets, the lower the SEP.

Additional comments

• I commend the authors for sharing the data and code.
• The authors site a set of works by Bateson, Gluckman, and Hanson to frame their hypotheses. The authors note the terminology of “silver spoon” and “match-mismatch” differs from epidemiological concepts, but it not clear why their developmental framework is preferred over the epidemiological one or even how exactly it differs (cf. Kuh, Lynch, and colleagues, http://jech.bmj.com/content/57/10/778).

---

## Round 0.2 · accepted · Accept

This was a very useful and entertaining exchange between authors and reviewers - and the responses are fully satisfactory. Well done - this is the way peer review should work (I take no credit for anything . . .) Congratulations - our field is impoverished as much for lack of these exchanges as raw data and new ideas.